# Framework and algorithms for identifying honest blocks in blockchain

**Xu Wang**[1,2,3], **Guohua Gan**[3,4,5], **Ling-Yun Wu**◉[1,2,3]*

**1** Key Laboratory of Management, Decision and Information Systems, Institute of Applied Mathematics, Academy of Mathematics and Systems Science, Chinese Academy of Sciences, Beijing, China, **2** School of Mathematical Sciences, University of Chinese Academy of Sciences, Beijing, China, **3** Laboratory of Big Data and Blockchain, National Center for Mathematics and Interdisciplinary Sciences, Chinese Academy of Sciences, Beijing, China, **4** Beijing Taiyiyun Technology Co., Ltd., Beijing, China, **5** University of Science & Technology Beijing, Beijing, China

* lywu@amss.an.cn

## Abstract

Blockchain technology gains more and more attention in the past decades and has been applied in many areas. The main bottleneck for the development and application of blockchain is its limited scalability. Blockchain with directed acyclic graph structure (BlockDAG) is proposed in order to alleviate the scalability problem. One of the key technical problems in BlockDAG is the identification of honest blocks which are very important for establishing a stable and invulnerable total order of all the blocks. The stability and security of BlockDAG largely depends on the precision of honest block identification. This paper presents a novel universal framework based on graph theory, called MaxCord, for identifying the honest blocks in BlockDAG. By introducing the concept of discord, the honest block identification is modelled as a generalized maximum independent set problem. Several algorithms are developed, including exact, greedy and iterative filtering algorithms. The extensive comparisons between proposed algorithms and the existing method were conducted on the simulated BlockDAG data to show that the proposed iterative filtering algorithm identifies the honest blocks both efficiently and effectively. The proposed MaxCord framework and algorithms can set the solid foundation for the BlockDAG technology.

## 1. Introduction

Blockchain is a decentralized transaction and data management technology which was firstly developed by Nakamoto [1] for Bitcoin. A sequence of blocks which contain block head and block body form blockchain chronologically. Marc Andreessen, the doyen of Silicon Valley's capitalists, listed blockchain as the most significant invention which has the potential to transform the world of finance and beyond since the Internet itself [2,3]. Nowadays blockchain has gained more and more attention, and has been applied in lots of areas, such as IoT [4–7], healthcare [8], finance [9–12] and supply chain [13]. But there still exist some developmental bottlenecks for blockchain. Swan [14] points out seven challenges that the blockchain technology faces, including throughput, security and so on. The throughput of current blockchain

**Data Availability Statement:** All relevant data are within the manuscript and its Supporting Information files.

**Funding:** LYW was supported by the Laboratory of Big Data and Blockchain, National Centerfor Mathematics and Interdisciplinary Sciences, Chinese Academy of Sciences. GG received salary

from Beijing Taiyiyun Technology Co., Ltd. The funders did not have any additional role in the study design, data collection and analysis, decision to publish, or preparation of the manuscript. The specific roles of these authors are articulated in the 'author contributions' section.

**Competing interests:** The authors have declared that no competing interests exist. The author GG is employed by Beijing Taiyiyun Technology Co., Ltd. This does not alter our adherence to PLOS ONE policies on sharing data and materials.

system under the Proof of Work (PoW) consensus mechanism is very low, nearly 6–7 transactions per second, while bank card such as Visa can process thousands of transactions per second. Many other consensus mechanisms such as Proof of Stake (PoS), Delegated Proof of Stake (DPoS), Proof of Importance (PoI), Proof of Luck (PoL), hybrid PoW/PoS, were proposed to accelerate the block validation process, but the performance of blockchain is still not substantially improved due to the occurrence of blockchain forks. Moreover, the consensus mechanisms with higher throughput often sacrifice the decentralization to some extent. The poor scalability limits the development and application of blockchain greatly, especially in the areas with high-frequency transactions.

Most of research on blockchain focused on improving blockchain from the perspective of privacy and security [15–19], and some worked on the consensus algorithms [20], while only a few of researchers conducted some research on its scalable limitation. Lin et al. [21] showed the SPV (Simplified Payment Verification) technology to handle with the scalable limitation of blockchain by only using the block head message without maintaining the full block information, which is equivalent to expand the block size in disguised form. Decker et al. [22] studied the information propagation in the Bitcoin network and pointed out that the propagation delay in the network is the main reason for blockchain forks, and the abandoned blocks due to blockchain forks are the source of low throughput and poor scalability. Biais et al. [23] modeled the blockchain protocol as a stochastic game to analyze the equilibrium strategies of miners. They found the longest chain without forks is a Markov perfect equilibrium and there also exist equilibria with forks, which leads to the orphaned blocks. The model can show how folks can be generated through information delays and software updates. Zohar et al. [24] proposed an alternative to the longest-chain rule used in Bitcoin, named as GHOST, which determines the main chain using all blocks in the subtree at the fork. By utilizing the abandoned blocks, GHOST can improve the security of blockchain, but the throughput remains the same. In another paper, Zohar et al. [25] presented the blockchain with directed acyclic graph structure (BlockDAG), which allows the blocks to reference multiple predecessors to incorporate the information from all blocks into log. The DAG structure works well and leads to an increased throughput. More and more researchers agreed that BlockDAG is the next generation direction for the blockchain technology. An increasing number of BlockDAG-based blockchain platforms appeared in industry, such as IOTA (Tangle), byteball, and XDAG [26–28].

The block ordering problem is the main concern under directed acyclic graph (DAG) structure. In order to avoid the double-spending attack (attackers try to spend the same cryptocurrencies more than once), the ordering of transactions is very necessary: People can determine the valid transactions through accepting the first transaction and rejecting the later one according to the order of two conflict transactions. And the ordering of transactions is dependent on the ordering of blocks. Unlike the traditional blockchain, the ordering of blocks in the DAG is not straightforward since the DAG is essentially a partial order graph. Therefore, many approaches were developed to derive a stable and invulnerable total order of blocks from the DAG. A typical approach for block ordering problem, as PHANTOM proposed by Zohar et al. [29], is mainly composed of two steps. First, distinguish the honest blocks (blocks generated and propagated timely by the miners who conform with the rules of blockchain) from the suspect dishonest blocks (blocks generated by the miners who deliberately keep the blocks secret for a long time or take other actions which are not in accordance with the rules of blockchain). Second, derive a full topological ordering based on the temporal information embedded in the topological structure of honest blocks. In the two-step approach, the quality of final block ordering is largely dependent on the precision of the honest block identification since the dishonest blocks might disturb the true temporal information. The more precise the honest block identification is, the better the block ordering.

In this paper, we presented a novel graph theory-based framework MaxCord for identifying the honest blocks in BlockDAG. Using this framework, several algorithms were developed and evaluated on the simulated BlockDAG data. Compared with the existing approach PHAN-TOM, the proposed algorithms are both effective and efficient which can lay the solid foundation for the following development of BlockDAG technology. The following of the paper is organized as follows. In Section 2, we described the honest block identification problem and proposed a novel general framework and illustrated its relationship with PHANTOM. Several algorithms for solving the problem were presented, including an iterative filtering algorithm MAXCORD-IFA. The algorithms were evaluated, analyzed and compared in Section 3 based on the simulated BlockDAG datasets. The conclusions were drawn, and further research directions were given in the last section.

## 2. Honest block identification problem

Briefly speaking, BlockDAG is a DAG of blocks, in which each block is constructed by one of the participated miners and linked to other blocks. The new block is connected to the existing DAG by referencing all tip blocks (those blocks are not referenced by any other blocks) of the DAG found by the miner. If there is no network delay, the DAG will become a directed chain where the first block (genesis block) is the root. Generally, the blocks constitute a growing DAG. If all participated miner honestly obey the rules of block generation and connection, the relative temporal order of blocks can be straightforwardly inferred, with acceptable small error which depends on the extent of network delay. However, the blocks generated by potential attackers might significantly disturb the temporal order information embedded in the Block-DAG. By analyzing the possible attack modes, we found that there are two key policies can be taken by the attackers:

1. keep their blocks secret and publish them later, e.g. after a certain transaction is confirmed;

2. only reference the specific blocks, e.g. the blocks created by themselves.

When generating a block and connecting it to the BlockDAG, the attackers might take any one or both two of the above policies depending on their attacking strategy. Whatever the policy the attackers use, the connection patterns between the blocks created by the attackers and the normal blocks are very different from the patterns within each group. Based on the intuition, we first defined a novel discord measurement which can estimate the possibility of two blocks belonging to different groups. Using the discord measurement, the honest block identification problem can be formulated as a maximum $k$-independent set problem.

### 2.1 Discord between blocks

Given a block DAG G = (V,E) where each vertex in V represents the block and each directed edge in E represents the reference link. For a block A, the future set is the blocks that can reach A, denoted as future(A,G). Similarly, the past set of block A is the blocks that can be reached from A, denoted as past(A,G). Fig 1 shows an example of the future and past sets. Naturally, there are still some blocks left except the block's future, past and itself. These blocks are the complementary set of the past(A,G), A and future(A,G), denoted as anticone(A,G):

$$anticone(A, G) = G\ (A \cup past(A, G) \cup future(A, G)).$$

The example of anticone is also illustrated in Fig 1. Normally, the anticone set denotes the blocks that are created during the block A's propagation time. When blocks are created very frequently, there are more blocks in the anticone set of each block.

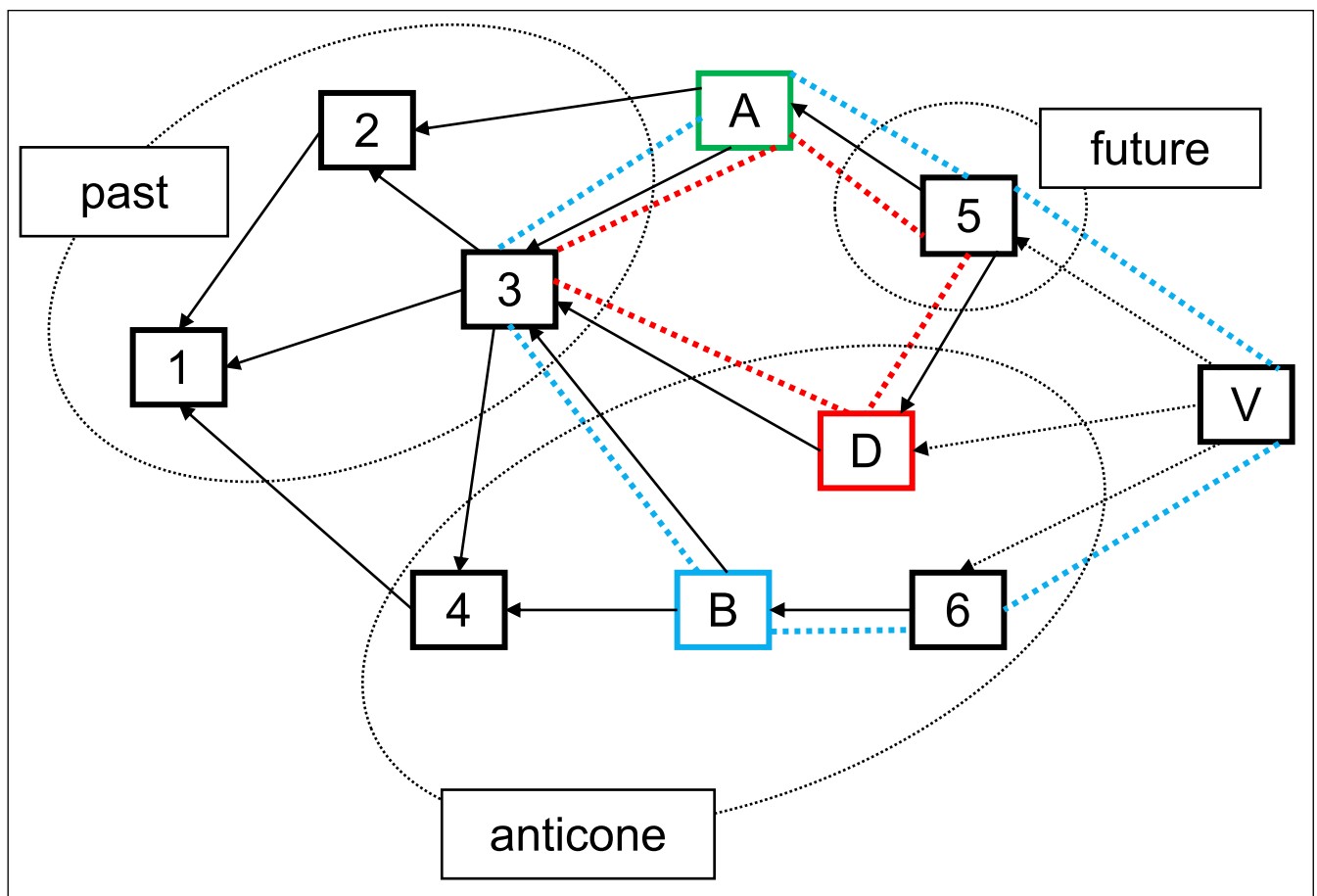

**Fig 1. The illustration of past, future, anticone of block A.** The discord between blocks A and B is 6 (the length of blue loop), and the discord between blocks A and D is 4 (the length of red loop).

We further define a novel measurement called discord to distinguish the blocks in the anticone set. For a block A, the discord between A and any block in past(A,G) and future(A, G), is defined as zero. The discord between A and a block B in anticone(A,G) is defined as the length of loop formed by A, B, their nearest common ancestor and nearest common descendant. For the situation where there is no common descendant between two blocks, a virtual block is added which references all tip blocks so that the blocks A, B, their nearest common ancestor and the virtual block can form a loop. An example of discord is illustrated in Fig 1.

Suppose $d_{ij}$ denotes the shortest distance from block i to block j in the original DAG, and $d_{AB}^A = \min_k(d_{Ak} + d_{Bk})$, $d_{AB}^D = \min_k(d_{kA} + d_{kB})$, $d_A^T = \min_{i \in Tips}(d_{iA})$, $d_B^T = \min_{i \in Tips}(d_{iB})$. The discord between blocks A and B can be mathematically defined by the following formula.

$$discord(A, B) = \begin{cases} 0 & if \ \min(d_{AB}, d_{BA}) < \infty \\ d_{AB}^A + d_{AB}^D & if \ d_{AB}^D < \infty \\ d_{AB}^A + d_A^T + d_B^T + 2 & if \ d_{AB}^D = \infty \end{cases} \qquad (1)$$

## 2.2 MaxCord framework

The discord is designed to estimate the temporal inconsistency (ambiguity) between two blocks. If two blocks are in the anticone of each other, their temporal order cannot be determined. But the time discrepancy of two blocks are bounded by their nearest common ancestor and nearest common descendant, since the real creation time of a block is bounded by its ancestors and descendants. The dishonest blocks always intend to hide or counterfeit its real creation time in order to carry out some attack such as double spending. Therefore, the discords between the dishonest block and most honest blocks should be very large, otherwise the real creation time of the dishonest block would be bounded by some honest blocks into a small interval. On the other hand, the discord of two honest blocks is normally much smaller. If the links between blocks are not artificially manipulated, the discord of two blocks should only depend on the network propagation speed and the block creation rate. When the network propagation speed or the block creation rate increases, the time discrepancy between the nearest common ancestor and the nearest common descendant will decline, however, the length of shortest path between two blocks will increase and cancel out the decline of time discrepancy to some extent. Therefore, the discords are not very sensitive to the network propagation speed and the block creation rate. The analysis on simulated BlockDAG datasets shows that the discords between two honest blocks are mostly smaller than 10 while the discords between the honest block and the dishonest block might be higher in two or more order of magnitude. Fig 2 shows the relationship between the block creation rate and the maximum discord between honest blocks. In every case of block creation rate, 100 simulations are conducted. The statistical analysis is shown in Table 1. Even when the block creation rate reaches 600 blocks per second, the discord between honest blocks still does not increase too much. Therefore, the discords can be utilized to filter the suspect dishonest blocks. In this section, we give a novel framework named MaxCord for identifying the honest blocks based on the discord.

Given a block DAG, we first calculate the discords for every pairs of blocks and get the discord matrix. Then the discord matrix is converted into a binary matrix where each element is 1 if the corresponding element in the discord matrix is larger than a preset threshold $d$ and 0 otherwise. By using the obtained binary matrix as the adjacency matrix, we can construct an undirected graph, in which each vertex represents a block. This graph is called $d$-discord graph of the given block DAG. Intuitively, if a block DAG only contains honest blocks, the degrees in the $d$-discord graph would be very small since the discords between most honest blocks are zero and the remaining non-zero discords are also very small. Considering the honest blocks are the majority, the honest block identification problem can be addressed by identifying the maximum subset of vertexes with small degrees.

Considering a graph G = (V,E), the $k$-independent set of G refers to the vertex subset V' in which the maximum degree in the induced subgraph does not exceed $k$. The maximum $k$-independent set problem is to find the $k$-independent set with maximum size, which is a generalization of classical maximum independent set problem. The maximum $k$-independent set can be formulated as the following integer programming, in which $x_s$ represents whether a certain vertex $s$ is selected and $a_{ij}$ denotes the element of adjacency matrix of the graph G.

$$
\begin{aligned}
\max \quad & \sum_i x_i \\
s.t. \quad & \sum_j a_{ij} x_i x_j \leq k, \forall i \\
& x_i \in \{0, 1\}, \forall i
\end{aligned}
\tag{2}
$$

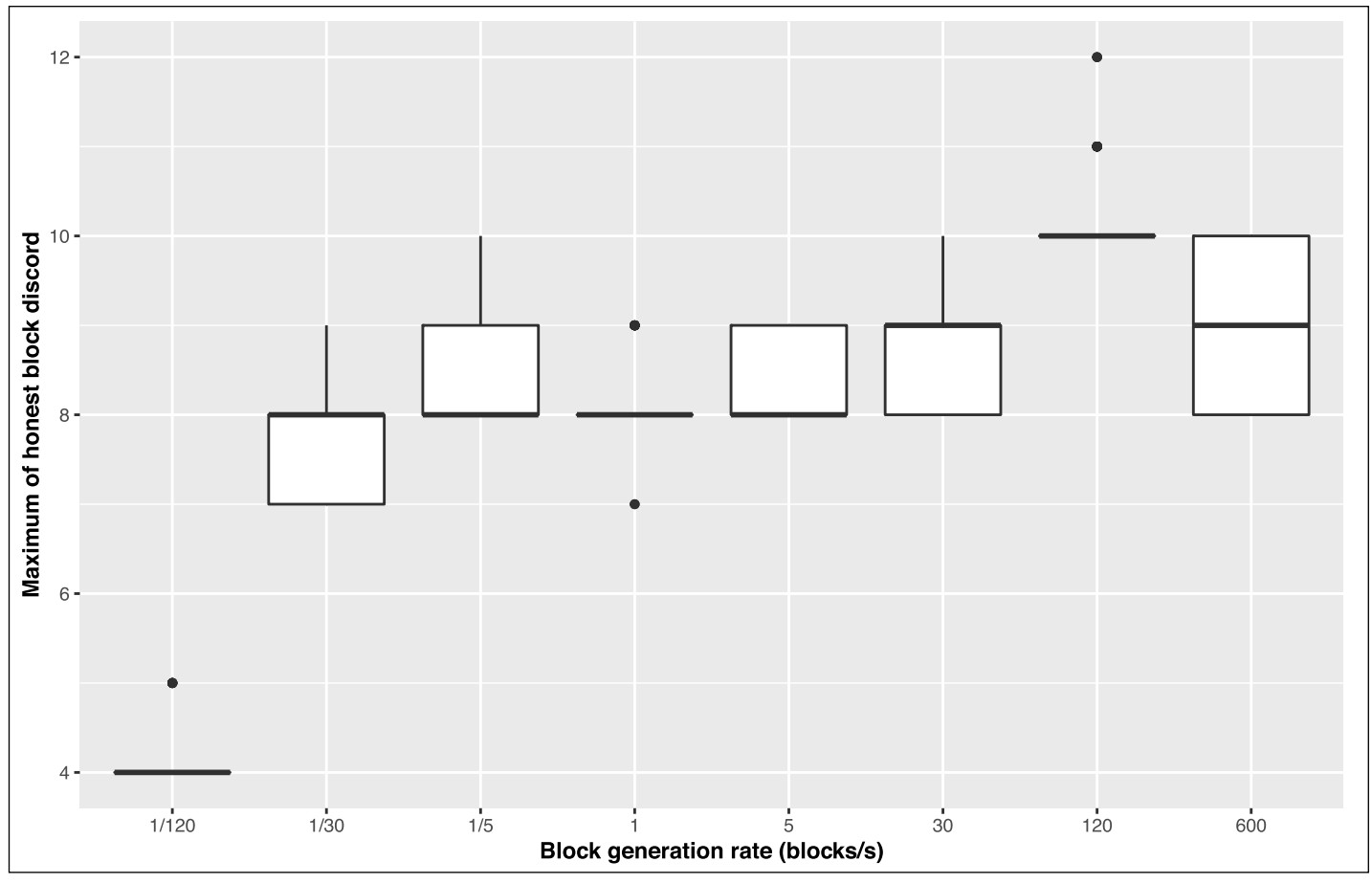

**Fig 2. The relationship between block generation rate and maximum honest block discord.**

Given the non-negative integer parameters $k$ and $d$, the honest block identification problem can be formulated as a maximum $k$-independent set problem in the $d$-discord graph of the given block DAG, denoted as MaxCord-$(k,d)$.

It is straightforward to verify that MaxCord-$(0,0)$ is equivalent to the longest-chain rule adopted in the traditional blockchain system such as Bitcoin. PHANTOM also falls into the MaxCord framework as it indeed solves the model MaxCord-$(k,0)$. PHANTOM's intuition is that the number of honest blocks in the anticone set of each honest block should be very small, e.g. less than $k$, since there are not many blocks generated during the propagation of the honest block. However, the parameter $k$ is closely related to the block creation rate and it is difficult to determine a proper value of $k$ in practice. Notice that the discord information is not fully utilized in the model MaxCord-$(k,0)$. In this paper, we focus on the other special case, i.e. MaxCord-$(0,d)$, which assumes the discords between each pair of honest blocks should be very small, e.g. less than $d$.

**Table 1. The statistical analysis of maximum honest block discords.**

| Block creation rate (blocks/s) | 1/120 | 1/30 | 1/5 | 1 | 5 | 30 | 120 | 600 |
|---|---|---|---|---|---|---|---|---|
| Mean | 4.08 | 7.68 | 8.50 | 8.09 | 8.34 | 8.78 | 10 | 9.07 |
| Maximum | 5 | 9 | 10 | 9 | 9 | 10 | 10 | 10 |
| Variance | 0.07 | 0.30 | 0.29 | 0.12 | 0.23 | 0.27 | 0.00 | 0.61 |

The parameter $d$ in the model MaxCord-$(0,d)$ plays a very important role as it determines the potential boundary between the honest blocks and the suspect dishonest blocks. Smaller threshold $d$ implies more stringent criterion of honest blocks. If the discords between a suspect dishonest block and all honest blocks are smaller than the threshold, the suspect dishonest block might be misidentified as an honest block. Therefore, larger threshold might decline the precision of honest block identification. On the other hand, if the discord between two honest blocks is larger than the threshold, one of these two blocks would be misidentified as suspect dishonest block. Thus, smaller threshold might decrease the recall. In a word, small $d$ emphasizes the precision of honest block identification, while large $d$ emphasizes the recall. This property will be exploited to develop a heuristic algorithm in the next section.

## 2.3 Algorithms for MaxCord-$(0,d)$

There are many algorithms of maximum independent set which can be directly applied to solve the model MaxCord-$(0,d)$, including approximation algorithms [30, 31] and exact algorithms for particular graphs [32, 33]. It is well-known that the maximum independent set problem is NP-hard, thus, there does not exist a polynomial-time exact algorithm. Generally, the exact algorithm is fast if the graph is very sparse, and very slow for dense graph. Besides the exact algorithm, a simple greedy algorithm was also implemented to solve the problem more efficiently, in which the vertex with the maximum degree is repeatedly removed as long as the remaining vertex set is not an independent set. The exact and greedy algorithms are named MAXCORD-EXACT and MAXCORD-GREEDY, respectively. In order to solve the honest block identification problem more efficiently, we further developed a new heuristic algorithm called MAXCORD-IFA.

Notice that when the threshold $d$ is large, the model MaxCord-$(0,d)$ has high recall. Large $d$ also leads to a sparse $d$-discord graph. By exploiting these two special characteristics of Max-Cord framework, MAXCORD-IFA solves the model MaxCord-$(0,d)$ by iteratively filtering out the suspect dishonest blocks identified by a series of models MaxCord-$(0,s)$ with $s > d$. Starting with a very large $s$, MAXCORD-IFA applies the exact algorithm of the maximum independent set problem to solve the model MaxCord-$(0,s)$, and removes the vertexes identified as suspect dishonest blocks. Then it decreases the value of $s$ and solves a new model MaxCord-$(0,s)$ defined on the remain blocks. The procedure is repeated until $s$ converges to the predetermined $d$. The detail of the MAXCORD-IFA algorithm is described in Fig 3.

There are two parameters in Algorithm 1. The parameter $d$ is the final value of the threshold for constructing the $d$-discord graph. In each iteration, the algorithm takes the top (1-alpha) percentage discords into account. That is, we gradually decrease the value of the threshold until it reaches the desired value $d$. In this way, the suspect dishonest blocks with high probability are filtered out step by step. The parameter alpha determines the sparsity of the $d$-discord graph in each iteration. If it is set close to 1, the $d$-discord graph becomes very sparse thus the computation of each iteration is fast, but more iterations are required to finish the algorithm. On the other hand, if it is set too low, the computation of each iteration would be unacceptably slow, and the result would be similar to that of the single iteration exact algorithm. Because MAXCORD-IFA is by means of multiple iterations while MAXCORD-EXACT is through one iteration to convert the discord matrix into binary matrix, as long as the parameters $d$ takes the same value, MAXCORD-EXACT is a special case of MAXCORD-IFA.

## 3. Results

In order to evaluate the proposed MaxCord framework, we applied several algorithms, including MAXCORD-EXACT, MAXCORD-GREEDY and MAXCORD-IFA to the simulated

| **Algorithm 1: MAXCORD-IFA** |
|---|

**Input:** the block DAG (n blocks), d, alpha

**Output:** the set of honest blocks

1: Begin

2:  Initialize matrix D(n*n)

3:  Initialize matrix M(n*n)

4:  For blocks i,j do

5:    D[i,j] ← discord(i,j)

6:  EndFor

7:  s ← quantile(D, alpha)

8:  While s > d do

9:    M ← zero matrix of the same dimension as D

10:   For i,j do

11:     If D[i,j] > s do

12:      M[i,j] ← 1

13:     EndIf

14:   EndFor

15:   h ← getMIS(M)        ►solve the maximum independent set problem

16:   D ← D[h, h]          ►extract the Rows and Columns h of matrix D

17:   s ← quantile(D, alpha)

18:  EndWhile

19:  M ← zero matrix of the same dimension as D

20:  For i,j do

21:    If D[i,j] > d do

22:     M[i,j] ← 1

23:    EndIf

24:  EndFor

25:  h ← getMIS(M)        ►solve the maximum independent set problem

26:  return h

27: End

**Fig 3. The pseudocode of MAXCORD-IFA algorithm.**

BlockDAG datasets and compared them with the existing method PHANTOM. Although there already exist some real BlockDAG systems, it is hard to get the real data. We simulated the BlockDAG datasets by using the probabilistic model of block creation and propagation on the P2P network. The computation power is assumed equally distributed, i.e. the blocks are created by the participated miners with equal probability. Two types of interval between two consecutively generated blocks is considered, including fixed interval and random interval (e.g. exponential distribution). The block transfer time on the P2P network is simulated using a gamma distribution. The parameters are tuned so that the throughput of blockchain with the longest-chain rule approximates the real world. The simulation algorithm is implemented in

the R package BlockSim which is available at http://github.com/wulingyun/BlockSim. The following simulation experiments were all run in R 3.5.1.

### 3.1 Parameter determination

The parameter $d$ of the MaxCord-(0,$d$) model is set identical for three algorithms MAXCORD-EAXCT, MAXCORD-GREEDY and MAXCORD-IFA. According to the analysis in the previous section, we take 8 as the default value of the parameter $d$ to identify the honest blocks in this study.

There is an additional parameter alpha in the algorithm MAXCORD-IFA. The parameter alpha is chosen not only to make sure the $d$-discord graph constructed in each iteration is sparse enough thus easy to solve by the exact algorithm of maximum independent set problem, but also to strike a balance between the number of iterations and the time taken by each iteration.

The influence of the parameter alpha on running time, precision and recall of different attack powers is shown in Fig 4. The results of MAXCORD-EXACT are also shown for comparison. Two attack modes are simulated, moderate attack (the attackers have 33% of all computation power) and heavy attack (the attackers have 49% of all computation power). In each attack mode, the algorithms are applied to 100 BlockDAG datasets. Each dataset contains 1000 blocks and the block creation rate is 10 blocks per second. The left part A of Fig 4 is the results under the moderate attack case while the right part B is the results under the heavy attack case. The horizontal axis represents that the parameter alpha ranges from 0.7 to 0.95. Notice that when the parameter alpha takes the value of 0.7, the results of MAXCORD-EXACT and MAXCORD-IFA are the same. Because when the value of alpha is small enough, MAXCORD-IFA takes only one iteration, and degenerates to MAXCORD-EXACT. From the aspect of running time, as the number of iterations increases, namely the gradual increment of the alpha, the total running time decreases although the decrements become less and less. Considering the indicators of precision and recall, 0.9 is the best choice for alpha no matter what level the attack power. In the following analysis of this study, we set the default value of alpha as 0.9.

### 3.2 Comparison between algorithms for MaxCord-(0,$d$)

We first evaluated several algorithms proposed for the MaxCord-(0,$d$) model, including MAXCORD-EXACT, MAXCORD-GREEDY and MAXCORD-IFA, and the comparisons among them are made. The BlockDAG is simulated with 1000 blocks and the block creation rate is 10 blocks per second. For each circumstance, we simulate 100 BlockDAG networks and the algorithms are applied to the same block DAG each time. The results are shown in Fig 5.

The horizontal axis represents the parameter $d$. With the increment of the parameter $d$, the indicator precision of the three algorithms decreases step by step, while their indicator of recall makes no big difference and approximates 1 when the attack power is moderate. The results of MAXCORD-GREEDY and MAXCORD-EXACT are almost identical under this case. MAXCORD-EXACT is a little bit better than MAXCORD-IFA in the aspect of the indicator of recall and they are nearly the same in the aspect of the indicator of precision. Stated in another way, MAXCORD-IFA improves the running time a lot by omitting a little bit honest blocks and maintaining the precision of the honest blocks when the attack power is moderate, but this little omission does not have large influence on the subsequent blocks ordering problem.

When the attack power is heavy, the parameter $d$ has larger influence on the results for MAXCORD-EXACT than the MAXCORD-IFA, that is MAXCORD-IFA is more robust. MAXCORD-IFA outperforms MAXCORD-EXACT no matter the indicators of precision or

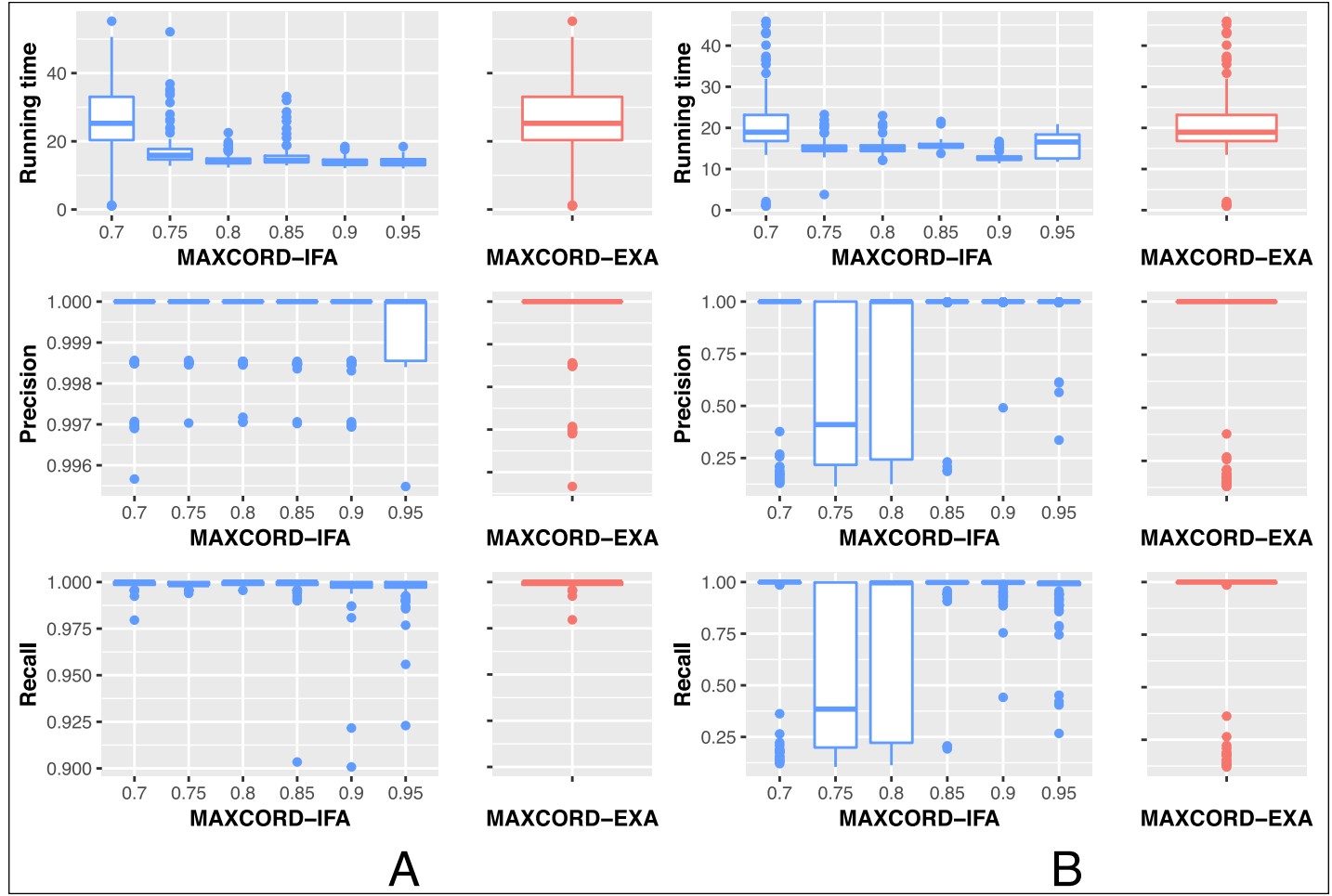

**Fig 4. The influence of the parameter alpha on MAXCORD-IFA.** Part A represents the case with moderate attack power, while part B represents the case with heavy attack power.

recall. MAXCORD-GREEDY is the worst under this case. In a word, MAXCORD-IFA is both effective and efficient in distinguishing the honest blocks from the suspect dishonest blocks.

### 3.3 Comparison between MAXCORD-IFA and PHANTOM

Because PHANTOM is a recursive algorithm which is very time-consuming, it is impracticable to apply PHANTOM in large-scales cases. Therefore, we only simulated small-scale cases to compare MAXCORD-IFA with PHANTOM. In detail, we simulated the BlockDAG with 200 blocks and the block creation rate is 1/10 blocks per second. The attackers' computation power percentage ranges from 0.1 to 0.45. For each attack circumstance, we conducted 10 simulations and both MAXCORD-IFA and PHANTOM are applied to the same 10 BlockDAG datasets. The parameter $k$ in PHANTOM takes the default value 3.

Fig 6 shows the precision and recall indicators of the two algorithms. It can be seen that MAXCORD-IFA can well recognize the honest blocks no matter how much computation power the attackers own. Even if the recall indicator of MAXCORD-IFA is not always 1 (very close to 1), which means it may omit a little bit honest blocks. This small sacrifice is worthwhile since it can save plenty of time. When the attack power is very strong, MAXCORD-IFA

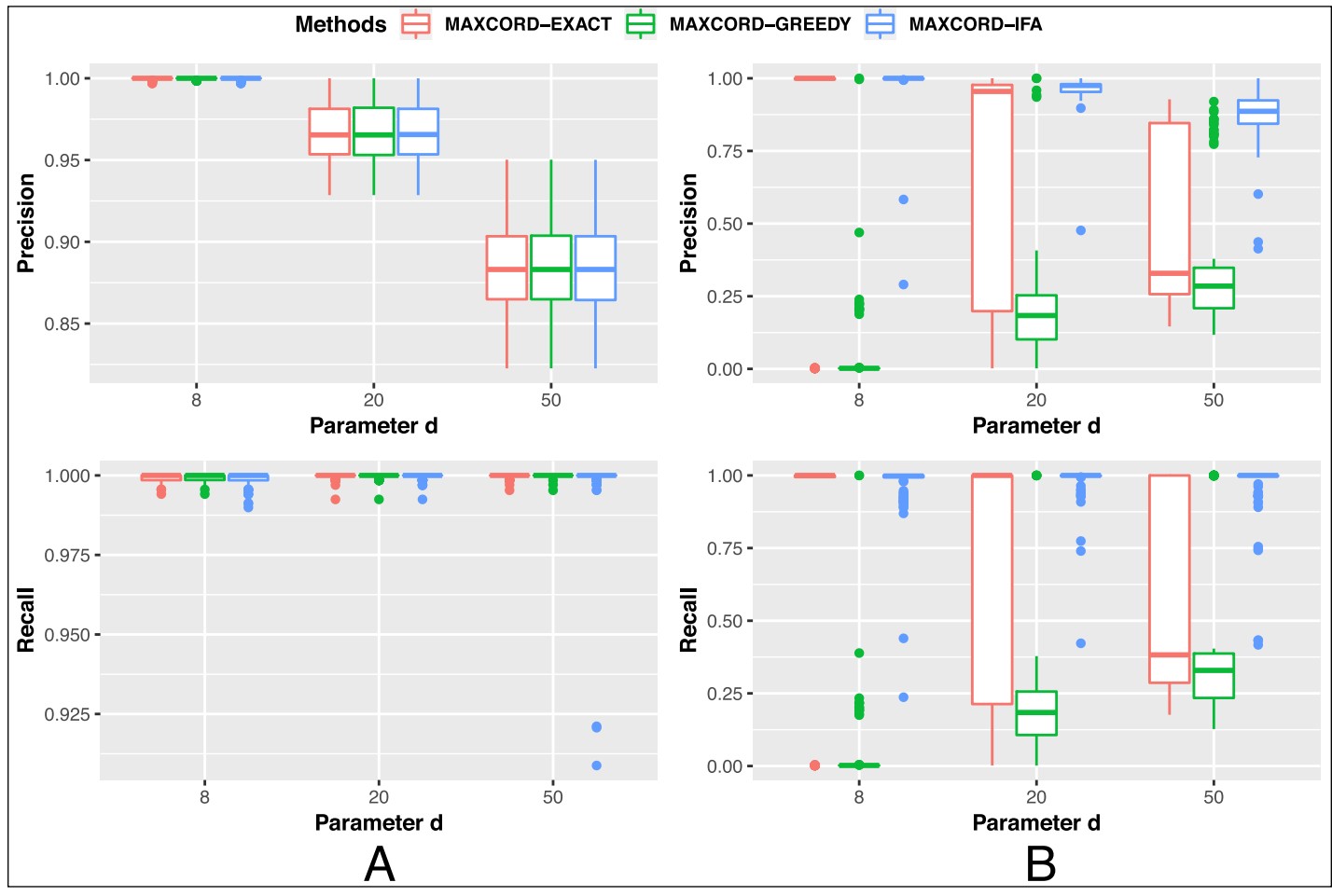

**Fig 5. The comparisons among the algorithms for MaxCord-(0,$d$).** The left part A represents the case with moderate attack power, while right part B represents the case with heavy attack power.

obviously outperforms PHANTOM. In a word, MAXCORD-IFA is an effective and efficient algorithm in the honest block identification problem.

## 4. Conclusions and discussions

In the paper, we introduce the honest block identification problem in the BlockDAG technology and present a novel universal framework of honest block identification problem through converting it into the maximum $k$-independent set problem on the basis of the definition of discord measurement between blocks. We point out the existing method PHANTOM is one of its special cases, and give several algorithms for the other special case MaxCord-(0,$d$), named MAXCORD-EXACT, MAXCORD-GREEDY, and MAXCORD-IFA.

Comparisons are made among these algorithms on the simulated BlockDAG datasets. MAXCORD-IFA outperforms PHANTOM to large extents, especially when the attack power is heavy. PHANTOM takes very long time therefore can only be applied to small-scale cases. The results of PHANTOM are sensitive to the value of parameter $k$ which is difficult to be determined for a given dataset. In large-scale cases, MAXCORD-IFA outperforms MAXCORD-EXACT and MAXCORD-GREEDY except that the recall of MAXCORD-IFA is a little bit lower than that of another two algorithms when the attack power is moderate. But this small

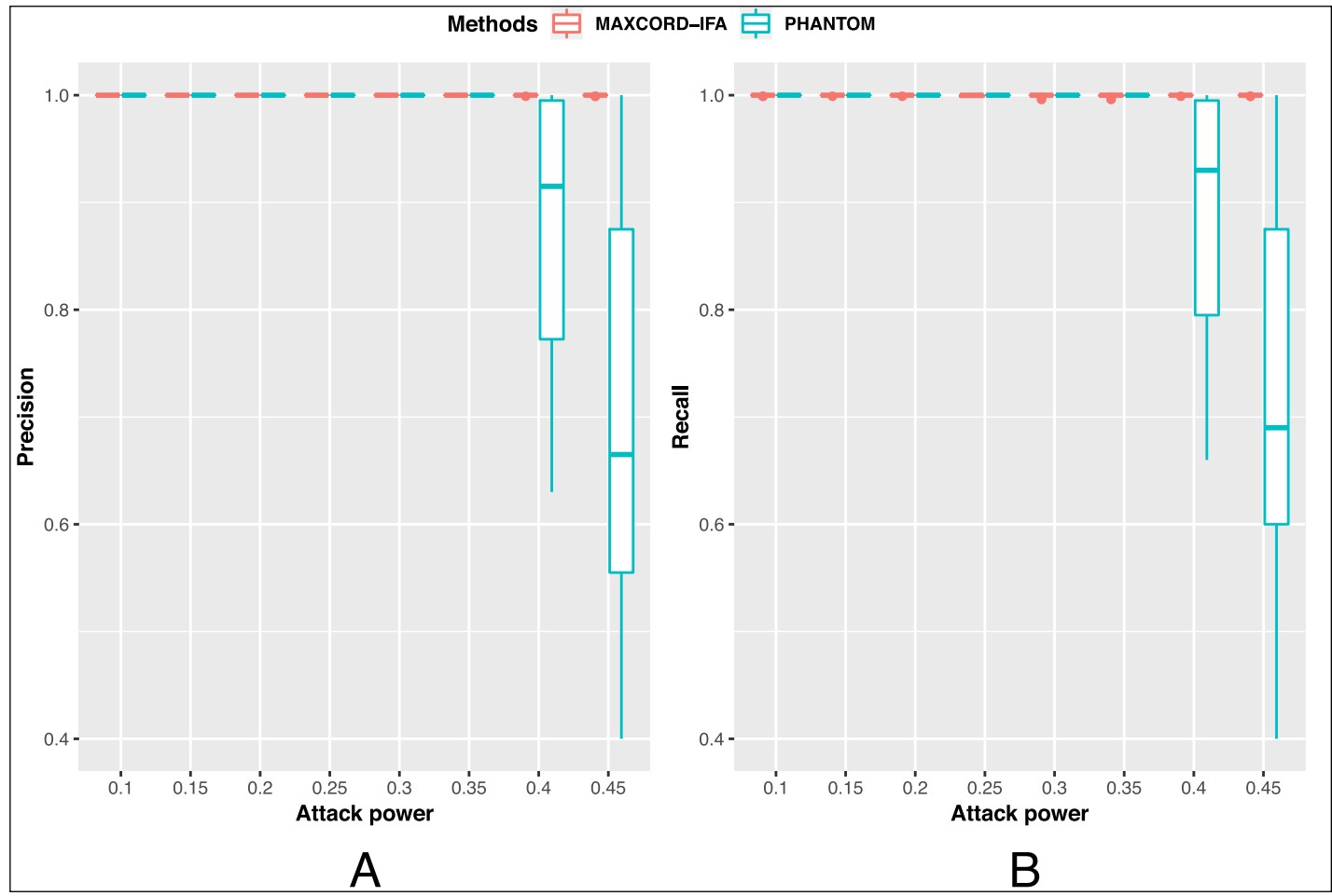

**Fig 6. The comparison between MAXCORD-IFA and PHANTOM.**

sacrifice, only omitting a little bit honest blocks while maintaining the precision of identification, can save lots of time and does not have significant influence on the subsequent block ordering problem.

This study is the first work on the MaxCord framework. In this direction, there are many problems deserved for future research. For example, MaxCord-$(0,d)$ and MaxCord$(k,0)$ are only two special cases of the MaxCord framework to identify the honest blocks. As our study suggests that MaxCord-$(0,d)$ is better than MaxCord-$(k,0)$, it is interesting to investigate the performance of general model MaxCord-$(k,d)$ for the cases where $d$ and $k$ are both non-zero. Development of better algorithms for MaxCord models is also a very important and challenging task. The influence of different algorithms on the following block ordering problem is still not very clear. The block ordering derived from the honest block sets identified by several algorithms might be very different, even if their honest block identification results are very similar. There are also other mathematical problems in the BlockDAG technology that is closely related to the honest block identification, such as the transaction fee allocation.

In this paper, we consider the BlockDAG in which the new block could be connected to all the tip blocks observed by the node when issuing the new block. All blocks are kept in the system and can be referenced directly or indirectly by the new blocks in the future. BlockDAG

attempts to identify the dishonest blocks by an algorithm independent of the construction of DAG. There also exist different approaches for the DAG-based blockchain technology. For example, IOTA Tangle attempts to distinguish the dishonest transactions from normal transactions by the tip selection algorithm based on MCMC (Markov Chain Monte Carlo) random walk and cumulative weights, and the dishonest transaction may fall into oblivion. It is interesting and important to study the pros and cons of two different approaches as well as the possible meld of them.

## Supporting information

**S1 Supporting Data. Simulated datasets and full results.**
(ZIP)

## Author Contributions

**Conceptualization:** Guohua Gan, Ling-Yun Wu.

**Investigation:** Xu Wang, Guohua Gan, Ling-Yun Wu.

**Methodology:** Xu Wang, Ling-Yun Wu.

**Software:** Xu Wang, Ling-Yun Wu.

**Supervision:** Ling-Yun Wu.

**Validation:** Xu Wang.

**Visualization:** Xu Wang.

**Writing – original draft:** Xu Wang.

**Writing – review & editing:** Guohua Gan, Ling-Yun Wu.

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
