## [Decision Letter · Decision Letter 0]

10 Oct 2019

PONE-D-19-24458

Framework and algorithms for identifying honest blocks in blockchain

PLOS ONE

Dear Dr. Wu,

Thank you for submitting your manuscript to PLOS ONE. After careful consideration, we feel that it has merit but does not fully meet PLOS ONE’s publication criteria as it currently stands. Therefore, we invite you to submit a revised version of the manuscript that addresses the points raised during the review process.

We would appreciate receiving your revised manuscript by Nov 24 2019 11:59PM. To enhance the reproducibility of your results, we recommend that if applicable you deposit your laboratory protocols in protocols.io, where a protocol can be assigned its own identifier (DOI) such that it can be cited independently in the future. For instructions see: http://journals.plos.org/plosone/s/submission-guidelines#loc-laboratory-protocols

We look forward to receiving your revised manuscript.

Kind regards,

He Debiao

Academic Editor

PLOS ONE

Journal Requirements:

1. Thank you for including your competing interests statement; "The authors have declared that no competing interests exist."

We note that one or more of the authors are employed by a commercial company:

 'Beijing Taiyiyun Technology Co., Ltd'.

Reviewers' comments:

Reviewer's Responses to Questions

**Comments to the Author**

1. Is the manuscript technically sound, and do the data support the conclusions?

Reviewer #1: Yes

Reviewer #2: Partly

2. Has the statistical analysis been performed appropriately and rigorously? 

Reviewer #1: N/A

Reviewer #2: I Don't Know

3. Have the authors made all data underlying the findings in their manuscript fully available?

Reviewer #1: Yes

Reviewer #2: No

4. Is the manuscript presented in an intelligible fashion and written in standard English?

Reviewer #1: Yes

Reviewer #2: No

5. Review Comments to the Author

Reviewer #1: It is a very interesting article where authors had the objective presents a novel universal framework MaxCord based on graph theory for identifying the honest blocks in BlockDAG. The authors suggest that their study is pioneering and new research needs to be done to develop a better model for MaxCord. This article is well written, results are considerable.

Reviewer #2: This paper present a new algorithm to recognize honest blocks in the blockchain technology. This problem is timely with a great importance. Using the generalization of max independent set problem seems very interesting. The English needs improvement. The following are my comments and suggestions if the paper is selected for publication:

-The authors correctly identified the scalability and low transaction rate as key challenges of blockchain. However, it only applies to BitCoin blockchain with proof of work (PoW) mining methods and there exist a lot of other mining methods such as Proof of Stake (PoS), hybrid PoW/PoS, Proof of Importance (PoI), Delegated Proof of Stake (DPoS), PoL (Proof of Luck), and Stellar Consensus Protocol (SCP) to accelerate the block validation process that is not mentioned in the introduction.

-It seems to me that the BlockDAG is somewhat related to the Tangle structure that is considered to (or at least is claimed to) replace the blockchain structure in the next generation of blockchains to solve the scalability issue. For example, see the following reference. Please clarify on this matter.

Popov, Serguei. "The tangle." cit. on (2016): 131.

-Please clarify on the “honesty” concept in the early pages of the paper. Is it equivalent to tampered blocks with false information, or blocks that are mined by illegitimate miner? The reader should wait until section 2 to get what you mean.

-Please explain Equation (1). Why discord is finite when $d_{AB}^D$ is infinite and there is no common descendent for blocks A and B. I may miss a point here but it does not mean that B is not in the anticone of A?

-Please explain why the discord is smaller than $10$ for two honest blocks. First, why it is larger for dishonest blocks and secondly where the threshold 10 coming from. Does it depend on the size of networks, transaction rates, etc?

-My another question is that this screening mechanism applies to blocks after they join the graph and anticones are formed. Due to the random block selection methods, it may take a long time for a pair of two blocks to appear in each other’s anticone. Please clarify.

-The difference between the Maxcord-IFA and Maxcord-0Greedy is not clear to me.

-It would be useful to introduce the simulation Environment.

-The figures are barely readable. Perhaps adding some Tables would be helpful.

-The authors compared Maxcord family algorithms, but I encourage the authors to compare their method against other methods as well.

6. PLOS authors have the option to publish the peer review history of their article (what does this mean?). If published, this will include your full peer review and any attached files.

Reviewer #1: No

Reviewer #2: Yes: Abolfazl Razi

---

## [Author Response · Author response to Decision Letter 0]

22 Nov 2019

We truly thank the editor and the reviewers for the opportunity and the constructive suggestions to further improve our manuscript. We have carefully revised the manuscript and made point-by-point responses to the comments. These comments are all valuable and very helpful for improving our work. We believe that all the concerns of the reviewers have been satisfactorily addressed. Please find our point-by-point responses to the comments in the file "Response to Reviewers".

---

## [Decision Letter · Decision Letter 1]

23 Dec 2019

Framework and algorithms for identifying honest blocks in blockchain

PONE-D-19-24458R1

Dear Dr. Wu,

We are pleased to inform you that your manuscript has been judged scientifically suitable for publication and will be formally accepted for publication once it complies with all outstanding technical requirements.

With kind regards,

He Debiao

Academic Editor

PLOS ONE

Additional Editor Comments (optional):

Reviewers' comments:

Reviewer's Responses to Questions

**Comments to the Author**

1. If the authors have adequately addressed your comments raised in a previous round of review and you feel that this manuscript is now acceptable for publication, you may indicate that here to bypass the “Comments to the Author” section, enter your conflict of interest statement in the “Confidential to Editor” section, and submit your "Accept" recommendation.

Reviewer #2: All comments have been addressed

Reviewer #3: All comments have been addressed

2. Is the manuscript technically sound, and do the data support the conclusions?

Reviewer #2: Yes

Reviewer #3: Yes

3. Has the statistical analysis been performed appropriately and rigorously? 

Reviewer #2: Yes

Reviewer #3: Yes

4. Have the authors made all data underlying the findings in their manuscript fully available?

Reviewer #2: Yes

Reviewer #3: Yes

5. Is the manuscript presented in an intelligible fashion and written in standard English?

Reviewer #2: No

Reviewer #3: Yes

6. Review Comments to the Author

Reviewer #2: Thanks you for addressing the comments. The paper is in much better shape now. The figures looks low-quality and blurred, I would only suggest to include high-quality figures in the last version, and further improve the English.

Reviewer #3: The manuscript makes important contributions to the research field and the authors adequately answered the questions of both reviewers. Based on that, I am in favor of its publication.

7. PLOS authors have the option to publish the peer review history of their article (what does this mean?). If published, this will include your full peer review and any attached files.

Reviewer #2: Yes: Abolfazl Razi

Reviewer #3: No

---

## [Editor Report · Acceptance letter]

31 Dec 2019

PONE-D-19-24458R1 

Framework and algorithms for identifying honest blocks in blockchain 

Dear Dr. Wu:

I am pleased to inform you that your manuscript has been deemed suitable for publication in PLOS ONE. Congratulations! Your manuscript is now with our production department. 

With kind regards,

on behalf of

Dr. He Debiao 

Academic Editor

PLOS ONE